# Energy Evaluation and Passive Damage Detection for Structural Health Monitoring in Aerospace Structures Using Machine Learning Models

**DOI:** 10.3390/s25164942

**Published:** 2025-08-10

**Authors:** Francesco Nicassio, Flavio Dipietrangelo, Antonella Gaspari, Gennaro Scarselli

**Affiliations:** 1Department of Engineering for Innovation, University of Salento, Via per Monteroni, 73100 Lecce, Italy; francesco.nicassio@unisalento.it (F.N.); flavio.dipietrangelo@unisalento.it (F.D.); 2Department of Mechanics, Mathematics & Management, Polytechnic University of Bari, Via Orabona 4, 70125 Bari, Italy; 3Department of Aeronautical and Astronautical Engineering, Boldrewood Innovation Campus, University of Southampton, Burgess Road, Southampton SO17 1BJ, UK; g.scarselli@soton.ac.uk

**Keywords:** Structural Health Monitoring, impact characterization, machine learning, artificial neural network, regression and classification approaches

## Abstract

Structural Health Monitoring (SHM) in aerospace engineering is more and more based on the use of Artificial Intelligence. In this manuscript machine learning algorithms were trained to identify and to characterize the structural effects of impacts on a typical aerospace aluminum panel. A significant experimental campaign was conducted to create suitable impact datasets (the vibrational behavior of the reinforced plate, acquired by piezo sensors). Shallow neural networks, properly trained, were applied to determine critical events affecting the operational conditions. The focus of the manuscript was double: on the severity of the event (a regression problem regarding impact energy) and on the detection of preexisting damage to monitored areas (a classification problem regarding the identification of damaged zones). The scope of this work was to demonstrate the validity of the machine learning approach as an SHM tool for impact effect characterization in a realistic aerospace structure (i.e., energy prediction with a percentage error never more than 10% and identification of previous damaged zones with an accuracy of more than 95%) and to demonstrate its computational efficiency despite the test complexity, provided that the selection of features is guided by a meaningful physical and mechanical interpretation of the underlying phenomena.

## 1. Introduction

Aluminum reinforced panels are still widely used in lightweight aerospace applications due to their high specific strengths and stiffnesses. In the service life of such panels, impacts are expected to arise from a variety of causes [1,2,3,4,5]. Debris can be projected onto the external surfaces at high velocities from the runway during aircraft take-offs and landings. Other examples include tools dropping on the structures during maintenance or collisions with small birds. Visual inspection can reveal a little damage on a plate, but significant damage may occur on the rivetted/bonded reinforcement area [6,7,8,9,10]. Reductions in structural stiffness and strength can occur and, consequently, propagate under further loading [11,12,13,14,15]. Thus, the behavior of these plates under impact has received increasing attention.

A proper structural monitoring system for aerospace applications has been proposed for the detection of low/intermediate energy impacts from dropped tools during operation and maintenance activities and high-energy impacts from strikes of foreign objects [16]. The response of thin aluminum plates to impact has been addressed in the technical literature by exploiting advanced analysis techniques, including artificial intelligence tools. Some studies refer to the variation in projectile parameters [17] (mass, nose shape, angle of incidence and hardness [18]) and targets (thickness, material, configuration and strength [19]). Boffa et al., in [20], demonstrated how a suitable combination of neural networks and Guided Wave-based methodologies was able to work as a monitoring tool for the localization of impacts with different energies and different excitation frequency contents. The results confirmed the good performance of the neural network and suggested a more extended experimental campaign aiming at defining the system precision, the possibility of reconstructing a fault and optimizing the data handling to reduce the computational cost. The authors of [21] presented a technique for impact event identification using vibration measurements retrieved from a single point of a structure. The localization problem was addressed by identifying the contributions of specific vibration modes as a signature of the impact location. The results demonstrated the method’s effectiveness in localizing impacts applied anywhere on the plate, as well as in rapidly estimating key load history parameters. The optimal balance between the accuracy and robustness of the method and between the amount of data and the time required for analysis remains a topic of research, especially when considering the application to more complex structures, i.e., in terms of material and geometry.

Data-driven machine learning models for Structural Health Monitoring (SHM) are not uncommon, but they are relatively new compared to traditional physics-based approaches. The momentum gained by machine learning in SHM is driven by its capability to extract accurate, adaptive and predictive insights from vibro-acoustic experimental data, outperforming conventional methods. This enhanced performance improves damage detection and prognosis, thereby leading to enhanced safety, reduced maintenance costs and optimized operational efficiency in aerospace applications. In recent years, their popularity has grown due to the increased availability of large datasets and advancements in algorithms. These models are highly valued in the technical literature for their ability to accurately predict and diagnose structural issues without relying on detailed physical models, as well as their effectiveness in handling complex patterns and nonlinearities in SHM data.

In this scenario, artificial intelligence models offer considerable promise for acoustic source localization [22,23,24]. Recent advancements in machine learning methods have opened new avenues for impact localization through Lamb wave characterization [25,26]. In [27], Ojha et al. proposed a framework based on probabilistic machine learning to identify impact locations, utilizing the wavelet scattering transform and Multi-Output Gaussian Process Regression. The first technique effectively extracted informative features from the captured Lamb waves, while the second one modelled the correlated spatial coordinates of the impact location and accounted for uncertainties within the data. Random forests and deep learning were adopted for training the source location models in [28] to automatically detect and localize impacts that may occur on aerospace structures during flight. In addition, the random forest model allowed the ranking of features. So, by deleting the least important features, the storage required to save the input and the computing time for the random forest were greatly reduced and an acceptable localization performance was still obtained. Although random forests can be effective for structured data and can be powerful due to their ability to handle interactions between features well, some advantages can be recognized in the use of neural networks, whose models are based on a less complex structure. In contrast, they require more careful tuning of parameters and optimization techniques.

The present work was carried out in the AeroSpace Structure Engineering Lab (AS.S.E. Lab) [29] and focuses on the monitoring of aluminum aerospace structures using machine learning approaches combining two methodologies previously developed in [25,26]. These works focused on the capability of advanced algorithms to localize and classify impacts in terms of their energy when they occur both on a monocoque aluminum plate and on a reinforced one.

The aim of the paper was to analyze the performance and integration capabilities of the above-mentioned monitoring tool, improving the approach by means of the implementation of a neural network for solving the regression problem of the impact energy. A further novelty is related to a passive monitoring technique based on the correlation between the experimental database and the vibrations due to impacts on previously damaged structures. The method was experimentally tested on a simple setup, namely, an isotropic reinforced aluminum flat panel. First, impact characterization was performed in terms of generated energy (i.e., impact velocity) in a range that guaranteed the replicability of impact events without changing the geometrical scenario (i.e., plastic deformations of the plate). Second, the guided waves related to the impacts were exploited: “free of charge” monitoring can be carried out using vibrational features with the scope of (i) classifying pristine or damaged structures and (ii) identifying damaged zones. Particular care was taken regarding the phenomenological aspects of the events by choosing convenient impact physical features and favoring algorithms that were able to let the SHM specialist be aware of the physical phenomenon behind the event. A robust approach is fundamental to its scalability towards more complex scenarios [30,31].

In the present paper, the materials used in the experimental setup (Section 2.1) and the methods (machine learning models and features in Section 2.2 and Section 2.3, respectively) are described. Then, in Section 3, the main results are discussed in terms of energy impact characterization (Section 3.1) and the “passive monitoring technique” (Section 3.2). Eventually, conclusions and an outline of future work close the work in Section 4.

## 2. Materials and Methods

### 2.1. Experimental Setup

The impact experiments were performed at the AS.S.E. Lab on an aluminum alloy (density: 2700 kg/m^3^, elastic modulus: 72 GPa, Poisson’s ratio: 0.33) flat reinforced plate. The dimensions (100 cm × 100 cm × 0.12 cm) of the plate and of the rivetted stiffeners (L shape of 1 cm × 1 cm × 0.12 cm), depicted in Figure 1, are typical for aeronautical applications (with Self-Plugging Mechanical Lock rivets in 5056-H14 aluminum alloy and a spacing of 5 cm). During the experimental tests, free–free conditions, i.e., the representative scenario of aerospace structures, were achieved using vibration-absorbing sponges under the entire sample.

Four circular PZT (Pb(ZrxTi1–x)O3) sensors of 10 mm PIC255—corresponding to the 600 series of the EN50324 European Standard [32]—were bonded onto the plate at the locations shown in Figure 2 by a solvent-free bicomponent glue made of a binder epoxy resin and hardener aliphatic amines. The square area surrounded by fours sensors was selected as the monitored zone. The experimental setup was focused on a “plate element” which is part of a complex structure (e.g., fuselage skin, wing surfaces, etc.) in an actual scenario (Figure 2a). In the case of a real “plate assembly”, there would be no quadrants outside the sensor array: each set of 4 PZT sensors would be able to monitor impact phenomena. A regular grid of 11 × 11 points provides 121 − 4 (sensors) = 117 impact locations, creating a symmetrical pattern, plus 50 impacts randomly chosen to reduce homogeneity and possible overfitting. Looking at Figure 2, the red dots, from A to U along the rows and from 0 to 20 along the columns, highlight the 167 impact positions. The responses from the sensors were acquired on a multi-channel acquisition system, a Picoscope 6402D, and post-processed with the relative oscilloscope software and MATLAB R2022b scripts. The following settings were used to acquire impact signals through Picoscope.
Oscilloscope voltage range: ±20 V;Timebase control: 5 ms/div;Sample control: 3.9 MHz/channel;Trigger: single mode, i.e., one of the four channels reaches an arbitrary fixed voltage threshold to capture all of the signals’ waveforms.

A server with a 4 GHz CPU frequency, 64 bits and 512 GB of RAM was used to train and test all presented machine learning models.

To generate the impact signal database, several experimental campaigns were conducted using a drop tower, simulating different scenarios for aircraft wing/fuselage bottom panels. The perpendicular impacts were generated by dropping a 10 g steel ball with a diameter of 5 mm. Since the drop-tower pipe had a diameter of 10 mm, in the worst scenario the maximum random deviation of the experimentally produced impact position would be 2.5 mm, which is manageable by the algorithms previously developed in [25,26].

For the impact energy characterization using a regression approach, all impacts occurred via free fall so that the impact energy could be easily calculated. The secondary bouncing effect was considered neglectable. The impact energies on the plate could be treated in terms of impact velocities, *v*, with the simplest and well-known gravitational potential energy of the ball ($v\;=\;\sqrt{2\mathrm{gh}}$) having a fixed *g* = 9.81 m/s^2^:*h_max_* = 0.34 m (*v* equal to about 2.6 m/s) to avoid permanent deformations of the plate and out-of-range signals of ±20 V.Four decremental heights (starting from *h_max_* with a 4 cm step and *v* equal to about 2.4, 2.2, 2.1 and 1.9 m/s).Subsequently, half and a quarter of *h_max_* were chosen (*v* equal to about 1.8 and 1.3 m/s, respectively), completing the dataset.

Smaller heights and relative velocities were not taken into account since they would have been far from practical interest with respect to actual aerospace scenarios.

The passive monitoring technique which uses the environmental vibrations due to impacts was developed with a classification approach. Five scenarios, i.e., five series of 167 impacts, played the leading role: impacts (i) on a pristine structure and (ii)–(v) on damaged structures in the quadrants α, β, γ and δ, as shown by the blue dotted squares in Figure 2a: three rivets were removed from a single quadrant, data were acquired and then the rivets were replaced before moving on to test another quadrant independently.

For metallic aerospace reinforced structures, one common type of damage is the kissing bond phenomenon between a plate and the one-dimensional reinforcing stiffener riveted to it: the artificial damage used in this research involved three missing rivets per quadrant (see the blue circles in Figure 2 representing the positions of the missing rivets).

### 2.2. Machine Learning Models

The impact energy prediction in a continuous domain can be considered a regression problem, in which the energy can be considered a function of the features calculated based on the measured quantities. Quite the opposite, damage detection may be considered a classification problem, in which the absence/presence of damage can identify patterns in the input data for each different scenario. Using machine learning algorithms, the goal is to define the most suitable features and functions that best fit the data. Simpler algorithms (e.g., linear/logistic regression, tree or nearest neighbor models) are not able to handle a dataset with homogeneous features in terms of characteristic differences between two or more neighboring impacts: adopting a supervised learning approach, as suggested in [25], a Shallow Neural Network (SNN) consisting of one or two hidden layers is considered a suitable choice. On the other hand, deep learning approaches require heavier computational effort as well as larger datasets.

In an SNN elementary neuron [33], each input is weighted and their sum with the relative bias forms the input to the transfer function. Neurons can use any differentiable transfer function to generate their output. In this work, the goals were to build (i) a regression model able to predict the impact energy as a continuous value in a defined range (Energy Regression Neural Network, ER-NN) and (ii) a pattern recognition model able to predict the existence of damage as a class among several possible scenarios (Monitoring Classification Neural Network, MC-NN). In the first case, the chosen network is made of hidden layers of sigmoid neurons followed by an output layer of linear neurons. Multiple layers of neurons with nonlinear transfer functions allow the network to learn nonlinear relationships between input and output vectors, while the linear output layer is used for the function fitting. On the other hand, with the pattern recognition problem, the network outputs must be constrained and the output layer should use a softmax transfer function (where the decision is made by the network).

Moreover, in these problems the main specification was to tolerate long computing times and high resource consumption to obtain precise results. For this reason, the Bayesian Regularization training algorithm was chosen because of its greater accuracy compared to the other main methods, such as Levenberg–Marquardt Backpropagation and Scaled Conjugate Gradient Descent. The process called Bayesian [34] provides a network training function that updates the weight and bias values according to Levenberg–Marquardt optimization. It minimizes a combination of squared errors and weights and then determines the correct combination to produce a network that is able to generalize well. Another specification is the generalization of models, which allows them to obtain the best predictions over test data without excessive correlation with training ones. The choice of a k-fold cross-validation procedure, considering 5 different combinations of training/test sets with a training/test ratio of 70/30 allowed such fitting.

### 2.3. Machine Learning Features

In the presented framework, the selection of indicators (i.e., features) for impact assessment is vital, considering the specific application requirements [35]. The analysis of a vibration signal expressed in terms of voltage in the time domain (PZT sensor output) was conducted by calculating various synthetic parameters which allow for the quantification of the level to be compared with any standardized reference levels [36,37,38]. The following ones were considered.

Peak, indicating the maximum positive or negative excursion of the vibration (see Equation (1)), suitable for short-duration impacts, like those of interest here.(1)$$x_{p}=\underset{i}{\max}\left|x_{i}\right|$$

Root Mean Square (RMS), i.e., the most significant measure of amplitude, as it takes into account the wave history over time and provides an amplitude value directly correlated with the energy content of the vibration (Equation (2)). Impulsive phenomena (i.e., impacts) exhibit significant maximum peaks, or the peak-to-peak values can be the indices most closely related to the presence of any damage.(2)$$x_{rms}=\sqrt{\frac{1}{N}\sum_{i=1}^{N}x_{i}^{2}}$$

Correlation coefficients reveal the strength and direction of the linear relationships between the values of two time series. They quantify the degree to which changes in one time series are associated with changes in another, providing a measure of their linear relationship. They are closely related to covariance, according to the following:(3)$$r=\frac{1}{N-1}\sum \left(\frac{x-\bar{x}}{\sigma_{x}}\right)\left(\frac{y-\bar{y}}{\sigma_{y}}\right)=\frac{\mathrm{cov}(x,y)}{\sigma_{x}\sigma_{y}}$$
where $\bar{x}$ and $\bar{y}$ and σ_x_ and σ_y_ are the means and standard deviations of two signals (e.g., x is related to the sensor signal acquired in healthy conditions, while y is related to the damaged structure). To properly compute the correlation coefficient highlighting significant changes in the peak signal, the signals must be synchronized so that they have the same staring time. The “alignsignal” function in Matlab was used to reach this goal.

Concerning the features in the frequency domain, the Short-Time Fourier Transform (STFT) represents a frequency content change over time by moving a window (e.g., Hamming, Hanning or Kaiser). A window multiplied by the signal allows the time-domain input signal to be divided into overlapping frames. Then, it is possible to apply the fast Fourier Transform to each obtained frame.

Power Spectral Density (PSD) gives evidence of how the power of a signal is distributed with a frequency. The PSD of a generic signal over time is expressed as the Fourier transform of its autocorrelation signal. The amplitudes of peaks in the frequency spectrum can indicate the presence and severity of damage.

Another typical parameter used to deal with guided waves is the Time of Flight (ToF). The ToF is derived from peak analysis in the Power Spectral Density domain. The ToF is identified by finding the time instant associated with the first relevant peak of the spectrogram, and it represents the time a wave takes to travel from a source to a sensor. In this work, a Hamming time-window was applied to the filtered vibrational signal—processed using a second-order high-pass Butterworth filter—for the computation of the Power Spectral Density (PSD). The characteristic wave propagation frequency of the A0 mode was set to 40 kHz, as reported in [25]. As already mentioned, this frequency was used to identify the corresponding arrival time in the spectrogram and to evaluate the Time of Flight (ToF).

Then, the ER-NN was trained, calculating the ToF, RMS, Energy and Max Peak for each of the 7 dropping ball heights.

Meanwhile, the MC-NN training for passive SHM was realized according to the following steps:For each of 5 experimental campaigns, 167 ToF, RMS, Energy, Max Peak and r values were calculated;For each of 4 sensors, the signals were aligned between pairs of signals, and subsequently the correlation coefficient was calculated for
Healthy (new campaign data) with healthy (previous publication data in [26]);Healthy with damaged in quadrant α;Healthy with damaged in quadrant β;Healthy with damaged in quadrant γ;Healthy with damaged in quadrant δ.

The working principles of the presented technique of “passive monitoring” using environmental vibrations due to impacts can be represented by the following flow chart (see Figure 3).

The following bullet list provides a set of practical guidelines that summarize the key steps of the approach illustrated in the flow chart.
Impact occurs.By using neural networks presented in [25] and in the present work, the impact is characterized in terms of its localization (the impact is also characterized in terms of its deviation from the predefined grid [26]) and energy (i.e., velocity).The closest training impact with respect to the one that just occurred is identified (the distance must be smaller than the training step grid).By the “signals alignment” command, the correlation between the two impact events is evaluated and passive monitoring of the structure is performed using the features ToF, RMS, Max Peak and correlation coefficient.

It is worth noting that the “passive” monitoring technique presented in this paper can effectively become an “active” technique by replacing random impacts with regular inspections, where an operator taps on the panel at set intervals to check structural integrity.

## 3. Results and Discussion

The following figures and tables present the several results of this work concerning impact energy regression prediction and passive structural monitoring using environmental vibrations due to impacts.

### 3.1. Impact Energy

In the first scenario, several energies were evaluated to create the entire database for the impact velocity characterization (see Table 1). It has to be pointed out that the ±10% tolerance range for the acceptable velocity prediction in Table 1 was selected as a practical empirical criterion based on preliminary analyses and supported by the consistency of the results. This range reflects a balance between prediction accuracy and inherent uncertainties in experimental conditions and modeling. Although not derived from a strict theoretical basis, it provides a reasonable baseline to evaluate the reliability of velocity predictions, and it can also be used for similar applications.

Regression algorithms were applied to the 167 impacts × 7 heights database using the features presented in Section 2.3.

The performances of the neural network were evaluated over the entire dataset with the following parameters (the best configurations in terms of prediction that resulted after several training cycles with different combinations of layer sizes):Bayesian Regularization training algorithm.Z-Score feature scaling (standardized dataset with zero mean and unit variance).Two layers with
Sigmoid transfer function in the hidden layers (20 neurons).Linear transfer function in the output layer (50 neurons).Maximum number of epochs (iterations): 1000.Random division of data (training set of 70% and test set of 30%).

Each network configuration was trained by the MATLAB train function, increasing the complexity of the model in terms of the number of neurons in the hidden layers. The best result was obtained by training the network as shown in Figure 4 (with four ToFs, four peaks and four RMSs per impact input from the time histories, such as that in Figure 5).

It is noteworthy that different impact energies have different maximum first peaks in the time history, without modifying the residual trend vibrations after a few milliseconds from the impact event.

In Figure 6, (i) the continuous lines represent the perfect correlation between target and predicted impact velocities, (ii) the dashed lines limit the acceptable predicted values (±10% of the target velocity) and eventually (iii) the scatter plots indicate the predicted 167 (training + test) impact velocities per target.

More detailed results can be analyzed by following Figure 7, which shows the distributions of predicted impact velocities for each dropping ball height (i.e., target impact velocities).

It can be seen in Table 2 and Figure 7 that the predicted values (blue bars and relative normal black distributions in the figure) and the actual velocities (continuous straight red lines) are well correlated: the percent error for all scenarios is never more than 10% of the mean predicted impact velocity (see the fourth column of Table 2). All normal distributions were verified by the Chi-squared test.

The following considerations can be drawn:For spaced impact velocities (i.e., h1, h6 and h7), all predictions can be easily identified by the algorithm, the distributions of values being clearly separated one from the other (see relative plots in Figure 7). The dropping heights of 0.34 m, 0.17 m, and 0.18 m are well-proportioned and evenly spaced relative to each other. Therefore, it was reasonable to expect a satisfactory outcome on the impact velocity prediction.Normalized errors of predicted velocity values with respect to the expected values are less than ±3%, which can be satisfactory, confirming the validity of the proposed approach; only one case was critical, corresponding to the height, h5, that almost overlapped with the immediately following condition, h6, which resulted in a difficult identification, due to a velocity difference of less than 3% between the two conditions.The previous consideration is supported by the low standard deviation values in Table 2, these being within the 10% limit set and showing a low dispersion of the predicted values around the relative mean values.In absolute terms, the number of impacts that resulted out of 10% bounds for each condition were less than one out of ten (see last column of Table 2).Examining the plots in Figure 7 reveals the presence of outliers and skewness in some distributions, suggesting that both normalized error and standard deviation are useful indicators for assessing the prediction quality when used jointly; notably, the critical case h5, with an error reaching approximately 7%, corresponds to a high number of out-of-band impacts, while case h3, with a normalized error below 3%, exhibits the highest standard deviation; both show the largest number of impacts outside the 10% bounds. Additionally, analysis of case h1 suggests lowering the 3% normalized error threshold for triggering a warning in the prediction quality to the 2.0–2.5% range.

### 3.2. Damage Detection

It is noteworthy that the monitoring scenario can be represented as a pattern recognition one, in which inputs are classified according to target classes. As an example, the impact occurred in one of the tested positions recorded by the same sensor is reported in Figure 8 as time histories—one for each damaged quadrant. In the time domain, the signal differences are so small, and the features evaluated (four ToFs, four peaks, four RMSs and four correlation coefficients per impact input) were necessary to correctly train the chosen algorithm.

Therefore, a feed-forward pattern recognition neural network was chosen, along with the Bayesian Regularization training algorithm, which provided the best precision in the previous predictions. The performances of the neural network in Figure 9 were evaluated over the entire dataset. For damage detection, the same 12 features were used as for the energy analysis, with the addition of the correlation coefficient—one per sensor, comparing the signals from the pristine and damaged plates—resulting in a total of 16 features.

The parameters used are reported in the following list:Z-Score feature scaling (data transformed to have zero mean and unit variance).Two layers with
(A)Sigmoid transfer function in the hidden layer (30 neurons);(B)Softmax transfer function in the output layer.Maximum number of epochs (iterations): 1000.Classification Error calculated based on an average of 50 test cases with the following random division of data (training set of 70% and test set of 30%).

The resulting confusion matrices in Figure 10 show satisfactory training and test results and an overall Classification Error of about 4%.

Only 35 test-case impacts (the entire dataset for the monitoring scenario included 835 impacts) provided wrong information concerning the status of the monitoring structure (equally distributed in terms of false-positive and false-negative values).

With an in-depth assessment of the “All Confusion Matrix” shown in Figure 10, the following conclusions can be drawn:For the pristine scenario, all actual 167 impacts provide the correct predicted class;For the plate with damage in quadrant α, 155 impacts provide the correct predicted class, 10 impacts provide predicted adjacent damaged areas (8 with damage in β, 2 in δ) and only 2 impacts provide misleading information;For the plate with damage in quadrant β, 157 impacts provide the correct predicted class, 9 impacts provide predicted adjacent damaged areas (2 with damage in α, 7 in γ) and only 1 impact provides misleading information;For the plate with damage in the quadrant γ, 156 impacts provide the correct predicted class, 9 impacts provide predicted adjacent damaged area (9 with damage in β) and only 2 impacts provide misleading information;For the plate with damage in quadrant δ, 165 impacts provide the correct predicted class, 1 impact provides a predicted adjacent damaged area (with damage in α) and only 1 impact provides misleading information.

Figure 11 shows the distribution of false positives for each damage scenario: the green circles represent the impacts that well predict the damaged area, while the red ones indicate wrong predicted impacts (the relative red Greek letters specify the wrong predictions). The worst result was obtained for the prediction of damages in quadrant β (see the predicted row test class “Damage in β” in Figure 10 and the β red circles in Figure 11): since the artificial damage was caused by manually unriveting the stiffener, probably, this less accurate result was a consequence of a more complex kissing bond in the quadrant β case.

Eventually, the relative ROC (Receiver Operating Characteristic) curves and the True-Positive Rates (TPRs) vs. the False-Positive Rate (FPRs) were provided as functions of the five investigation classes (one pristine and four damaged plates). The training ROCs were close to the point in the upper-left corner of the space, representing about 100% sensitivity (equal to the True-Positive Rate) and 100% specificity (equal to 1–the False-Positive Rate), with just two false negatives in the training phase. It is noteworthy to highlight, also, that the test ROCs were all far above the diagonal line (the so-called “line of no-discrimination” from the bottom-left to the top-right corners of the space). The overall (training + test) ROCs can be used to measure the accuracy of the presented technique: the Areas Under the Curve (AUCs) were always bigger than 0.9 (AUC_N.D._ = 0.99, AUC_α_ = 0.98, AUC_β_ = 0.97, AUC_γ_ = 0.99, AUC_δ_ = 0.99), as shown in Figure 12, certifying the excellent developed classification method and the resulting outstanding passive monitoring technique.

In conclusion, the deep aim of this approach was the maximization of the efficiency (in terms of accuracy for the predicted impact energy and the precision of the monitoring results) of the implemented neural networks. The isotropic aluminum reinforced panel was a relatively simple item on which the machine learning algorithm was trained and tested to collect proper know-how for more complex future applications.

## 4. Conclusions

This article presents a novel impact prediction tool for traditional aerospace thin reinforced structures. The main goals were both to characterize energy events in terms of impact velocity and to monitor structural health in a passive manner. The experimental campaigns provided information about the vibrational behavior of plates due to impacts at different energies and/or structural health conditions. By using different features (Time of Flight, Root Mean Square, Energy and Max Peak) and shallow neural networks, excellent agreements between training/test results and actual data were found in terms of impact velocity prediction and passive monitoring. Specifically, for the regression impact energy problem, only 101 impacts out of 1169 were wrongly predicted (actual velocity vs. evaluated velocity, with an error bigger than 10%). The merit of this approach was demonstrated by comparing the proposed regression outcomes with the classification results in [26]. For the classification monitoring problem, the network provides the correct scenario (pristine structure or damage in quadrants α, β, γ and δ) with an accuracy of about 96%.

Limitations remain regarding the impact magnitude (energy), which are primarily due to the performance constraints of the acquisition system in avoiding signal saturation. Further improvements are expected from the use of a more repeatable impact positioning system, as well as in the procedures for rivet removal and the introduction of artificial damage. Realistic scenarios could also be considered by analyzing industrial plates or in-field specimens.

Despite some open points, criteria for evaluating the effectiveness of this methodology have been identified, which are primarily related to the following aspects:Computational efficiency (less than five minutes for the training of each neural network and a few seconds for predicting impact energy and the passive monitoring phase);The physical significance of the selected indicators;The repeatability and reproducibility of results, even with varying experimental conditions (impact energy and localization)

There are several features that make the presented approach interesting. First, the shallow neural networks based on Bayesian Regularization that are employed are relatively simple, using the main basic features of impacts. Nets like the ones presented in this manuscript can be implemented in any programming language (Matlab was used here, but other open-source software may also be considered) and successively run with limited resource consumption. Also, the cost-effectiveness of the proposed method (less than five minutes for the net training and a few seconds for test-case prediction) makes it particularly interesting for future works. Composite applications, curved geometries and experimental setups with embedded sensors will be utilized to verify the robustness of the presented models. Moreover, different approaches, such as hybrid machine learning models or data augmentation techniques, will be tested for further applications (i.e., setup miniaturizing for drone scenarios). Eventually, the previous aspects will be analyzed so that the approach will be more closely aligned with practical applications, such as determining the influence of the incidence angle between the plate and the debris, and a wider range of energy levels and a reduced number of sensors will be implemented. Further studies will address measurement uncertainties and their influence on velocity predictions; some of these activities are already in progress, though they are beyond the scope of this work. In particular, an interlaboratory comparison is being carried out to evaluate the consistency of the approach across different setups and to identify potential sources of variability, thereby enhancing its overall robustness and reliability.

## Figures and Tables

**Figure 1 sensors-25-04942-f001:**
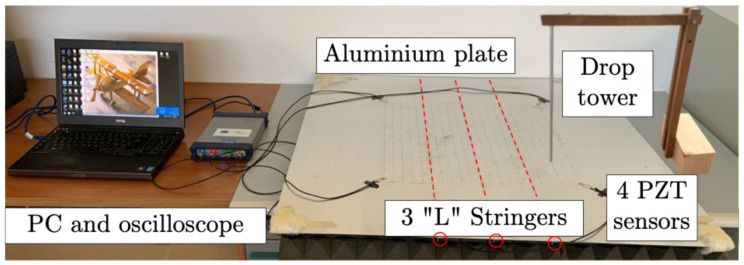
Experimental setup.

**Figure 2 sensors-25-04942-f002:**
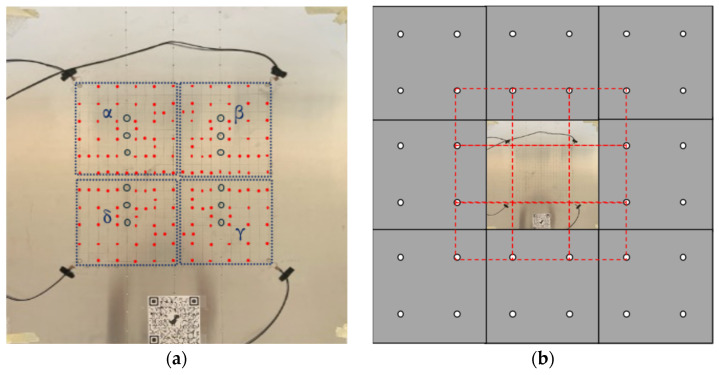
Plate scenarios: (**a**) 167 impacts (red circles), removed rivets (blue circles) and 4 quadrants (dotted blue squares); (**b**) “plate assembly” with no corners outside the sensor array. The red dotted lines identify the working area depicted in (**a**).

**Figure 3 sensors-25-04942-f003:**
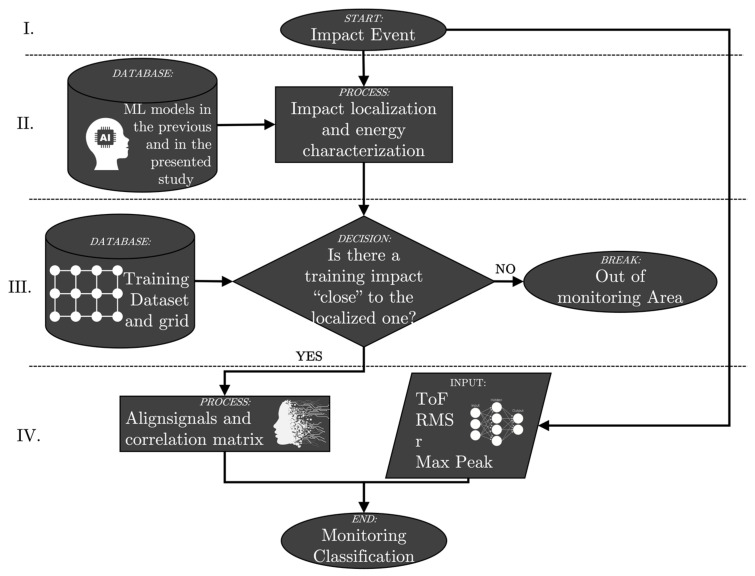
Flow chart for passive monitoring process. Previous ML models are described in [25].

**Figure 4 sensors-25-04942-f004:**
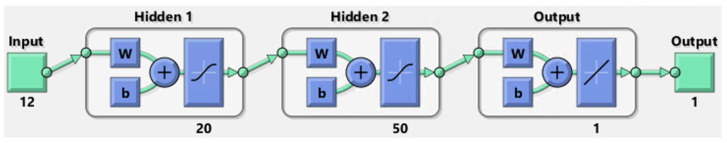
Flow chart for passive monitoring process (to solve the impact energy recognition problem): green path and blue steps inside each hidden layer are highlighted. The weight and bias of the neurons are indicated as ‘w’ and ‘b’, respectively.

**Figure 5 sensors-25-04942-f005:**
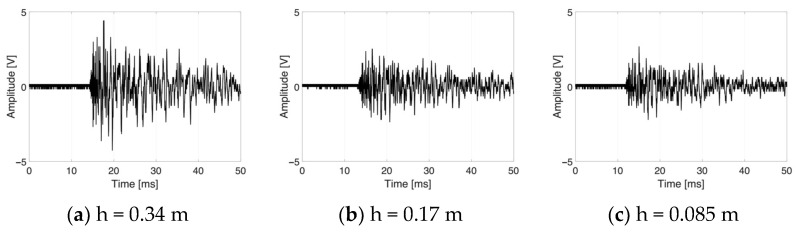
Signals received from the same transducer (top-left) under the same impact location (Q14) with different impact energies.

**Figure 6 sensors-25-04942-f006:**
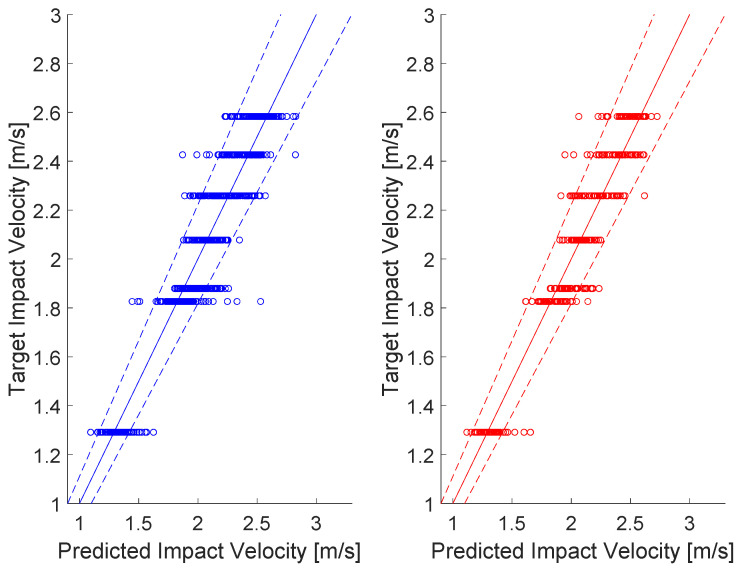
Predicted vs. actual impact velocity plots (left: training results in blue, right: test results in red).

**Figure 7 sensors-25-04942-f007:**
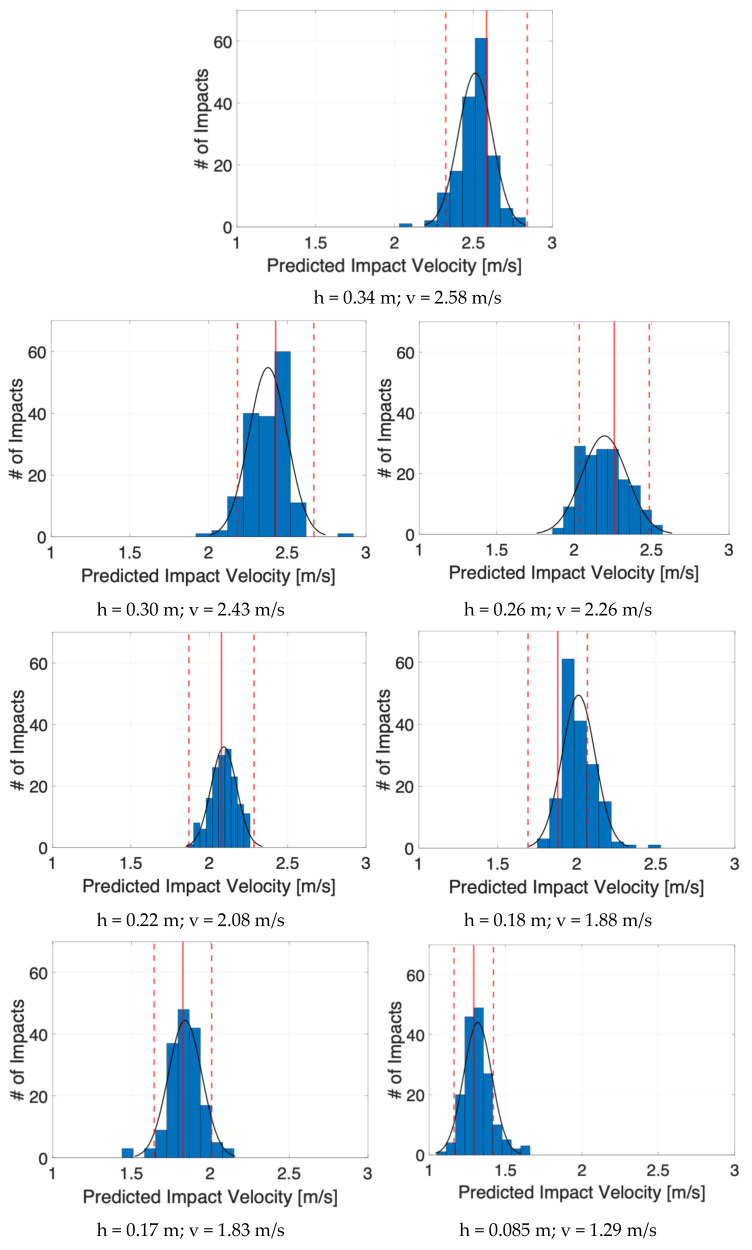
Distribution of predicted velocities for all heights (blue bins with relative normal distributions in black). Continuous red lines indicate the target velocities, the dashed ones the acceptable limits. # of impacts on the y-axis label stands for the number of impacts.

**Figure 8 sensors-25-04942-f008:**
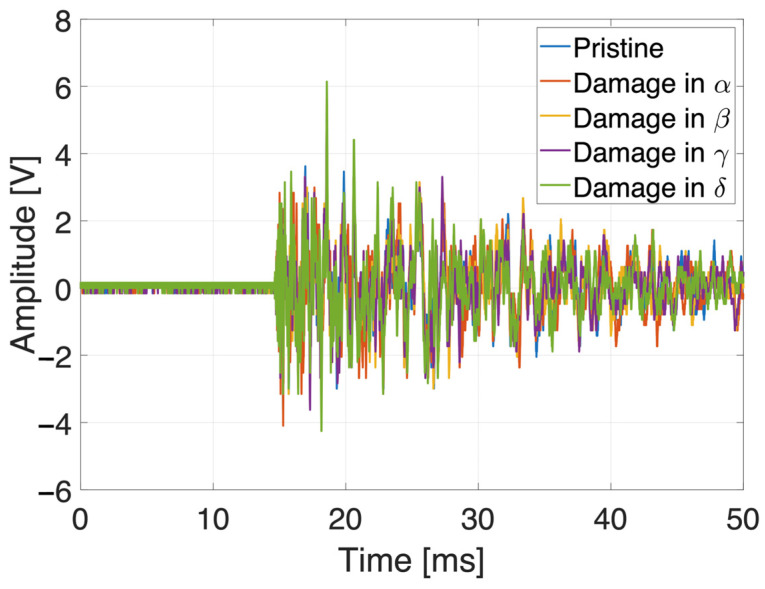
Signals received from the same transducer (top-left) under the same impact location (K7) with the same impact energy (h_max_) and a different damage quadrant.

**Figure 9 sensors-25-04942-f009:**
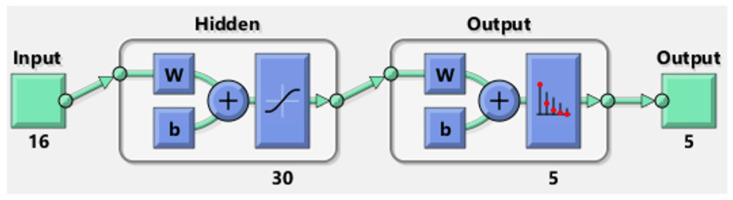
MATLAB representation of the neural network used for pattern recognition, with two layers, a sigmoid transfer function in the hidden layer and a softmax transfer function in the output layer: green path and blue steps inside each hidden layer are highlighted. The weigh and bias of the neurons are indicated as ‘w’ and ‘b’, respectively.

**Figure 10 sensors-25-04942-f010:**
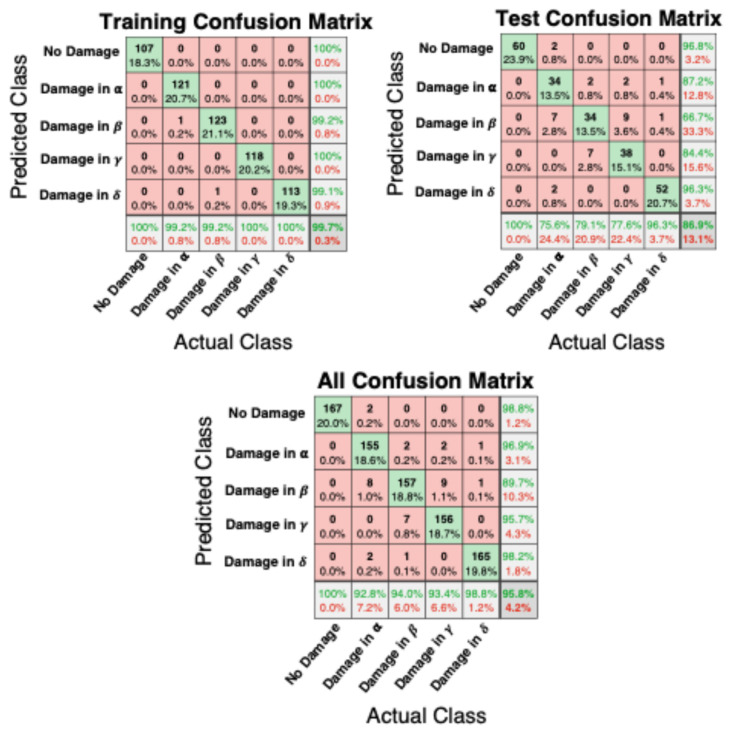
Training, testing and all confusion matrices. The five damage classes considered are reported on both axes of each matrix. Green cells on the diagonal indicate the correct predictions, red cells the wrong ones.

**Figure 11 sensors-25-04942-f011:**
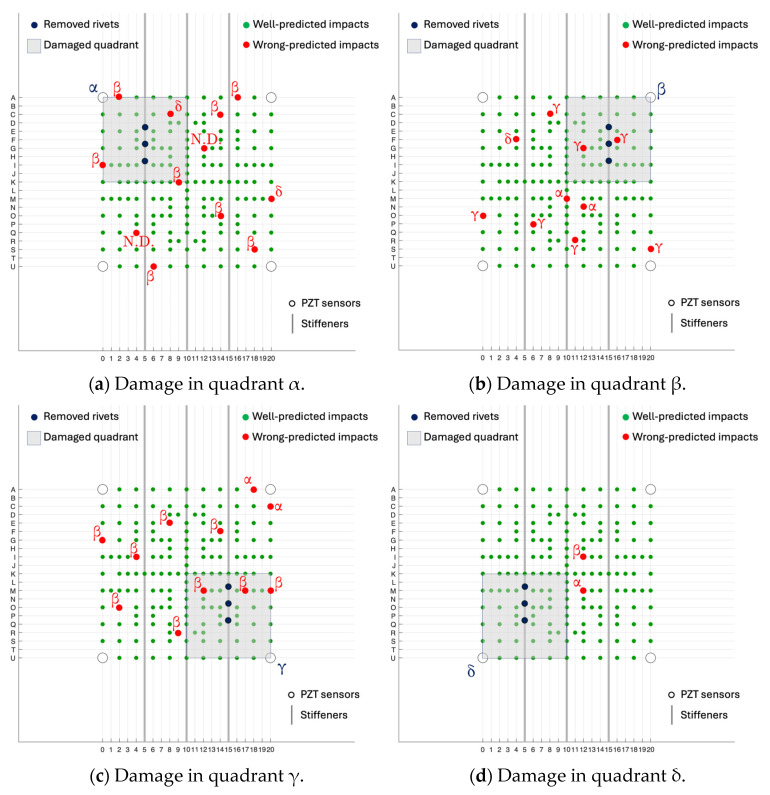
Test case results for the 4 damage scenarios. N.D. stands for ‘No Damage’ class.

**Figure 12 sensors-25-04942-f012:**
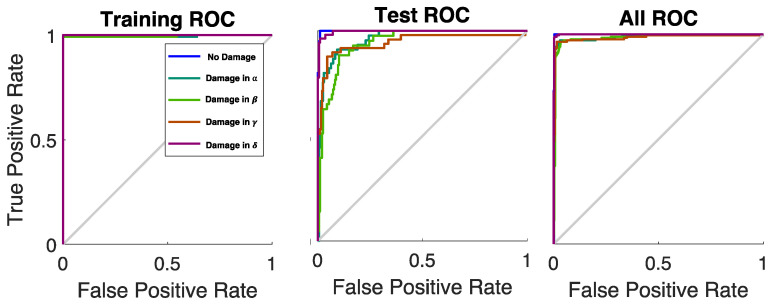
ROC curves for each scenario (see legend: no damage: blue, damage in α: dark green, damage in β: light green, damage in γ: brow, damage in δ: violet). The gray bisector line marks where True Positive Rate equals False Positive Rate, indicating no discrimination, i.e., random guessing.

**Table 1 sensors-25-04942-t001:** Impact features: heights (h), velocities (v) and acceptable limits on the velocity prediction.

ID	Dropping Ball Height [m]	Impact Velocity [m/s]	Acceptable Velocity Prediction Limits	
			Lower Limit (−10%) [m/s]	Upper Limit (+10%) [m/s]
h1	0.34	2.58	2.32	2.84
h2	0.30	2.43	2.18	2.67
h3	0.26	2.26	2.03	2.48
h4	0.22	2.08	1.87	2.29
h5	0.18	1.88	1.69	2.07
h6	0.17	1.83	1.64	2.01
h7	0.085	1.29	1.16	1.42

**Table 2 sensors-25-04942-t002:** Impact features: heights, velocities and acceptable prediction limits. Regression results for each impact velocity case.

ID	Actual Impact Velocity [m/s]	Predicted Impact Velocity		Error: Predicted vs. Actual Ratio [%]	Out of 10% Bounds [# of Impacts]
		Mean [m/s]	Standard Deviation [m/s]		
h1	2.58	2.51	0.11	−2.76	5
h2	2.43	2.38	0.12	−1.99	11
h3	2.26	2.20	0.15	−2.80	20
h4	2.08	2.09	0.08	0.75	1
h5	1.88	2.01	0.11	6.91	36
h6	1.83	1.84	0.11	0.77	11
h7	1.29	1.32	0.09	2.05	17

## Data Availability

The raw data supporting the conclusions of this article will be made available by the authors on request.

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
