# Peer review of "Energy Evaluation and Passive Damage Detection for Structural Health Monitoring in Aerospace Structures Using Machine Learning Models"

_sensors, 2025, doi:10.3390/s25164942_

Round 1
Reviewer 1 Report
Comments and Suggestions for Authors
The manuscript presents a comprehensive study on the application of machine learning models for structural health monitoring in aerospace structures, focusing on impact energy characterization and passive damage detection. The experimental setup, methodology, and results are well-documented, and the findings demonstrate promising accuracy in both regression and classification tasks. Overall, the topic of this research is interesting, and the manuscript is well structured. The detailed comments are given as follows.
- This study mentions a grid of 11×11 points for impact locations, resulting in 117 impacts plus 50 randomly chosen impacts. What criteria were used to select the additional 50 random impact locations, and how was their distribution ensured to avoid bias in the dataset?
- The authors use seven dropping heights for impact energy characterization. Were any preliminary tests conducted to determine the range of heights, and how was this range justified in terms of practical aerospace scenarios?
- Four PZT sensors were bonded to the plate. How was the placement of these sensors optimized to ensure maximum coverage and sensitivity for impact detection, especially given the plate's dimensions?
- The correlation coefficient is used to compare signals from healthy and damaged structures. How was the synchronization of signals achieved, and what challenges were encountered during this process?
- The shallow neural network architecture includes 20 neurons in hidden layers for regression and 30 for classification. What was the rationale behind choosing these specific numbers of neurons, and were other configurations tested?
- A 70/30 split was used for training and testing. Was cross-validation employed to ensure the robustness of the model, and if not, what measures were taken to prevent overfitting?
- The authors highlight the use of Bayesian Regularization for training. How does this method compare in terms of computational efficiency and accuracy to other training algorithms like Levenberg-Marquardt or Scaled Conjugate Gradient?
- Broaden literature review on ML-based damage detection. E.g., Ground penetrating radar-based automated defect identification of bridge decks: A hybrid approach.
- The manuscript claims the method is computationally efficient. Can the authors provide specific metrics such as training time and prediction time to support this claim, especially for real-time applications?
- More future research should be included in conclusion part.
Author Response
We thank the reviewer for the constructive and encouraging comments. Please find below our point-by-point responses. In the revised manuscript, the changes are highlighted in yellow for easier verification of the implemented adjustments.
Q1.0 - The manuscript presents a comprehensive study on the application of machine learning models for structural health monitoring in aerospace structures, focusing on impact energy characterization and passive damage detection. The experimental setup, methodology, and results are well-documented, and the findings demonstrate promising accuracy in both regression and classification tasks. Overall, the topic of this research is interesting, and the manuscript is well structured. The detailed comments are given as follows.
A1.0 - We are grateful to the reviewer for the valuable comments and insightful suggestions, which have helped us to improve the quality of the manuscript, according to the list reported below.
Q1.1 - This study mentions a grid of 11×11 points for impact locations, resulting in 117 impacts plus 50 randomly chosen impacts. What criteria were used to select the additional 50 random impact locations, and how was their distribution ensured to avoid bias in the dataset?
A1.1 - We thank the reviewer for this relevant observation. In response, we performed a denser sampling in the central area, as this region tends to exhibit a lower signal-to-noise ratio, likely due to its increased distance from the transducers.
Q1.2 - The authors use seven dropping heights for impact energy characterization. Were any preliminary tests conducted to determine the range of heights, and how was this range justified in terms of practical aerospace scenarios?
A1.2 - We thank the reviewer for the helpful remark and for pointing out the relevant threshold constraints. As suggested, the minimum height was set at 8 cm to remain above the lower threshold, while the maximum was limited to 0.34 m in order to avoid exceeding the upper limit. These considerations were taken into account in the revised version of Section 2.1.
Q1.3 - Four PZT sensors were bonded to the plate. How was the placement of these sensors optimized to ensure maximum coverage and sensitivity for impact detection, especially given the plate's dimensions?
A1.3 - The proposed approach is focused on one single “plate element” which is, commonly, part of a complex structure (e.g., fuselage skin, wing surfaces…) in a real scenario. As the Reviewer can see in the renewed and ameliorated Figure 2, in the case of a real “plate assembly” there would be no corners outside the sensors array: each set of 4 PZT (dotted squares in Figure 2) would be able to monitor impact phenomena. For taking into a proper account the influence of boundary conditions (joints between each plate element) an interesting follow-up study has already been launched. This is now explicitly stated in the revised manuscript as a motivation for the choice of the presented experimental setup.
Q1.4 - The correlation coefficient is used to compare signals from healthy and damaged structures. How was the synchronization of signals achieved, and what challenges were encountered during this process?
A1.4 - We thank the reviewer for the comment. As noted in lines 233–235, we used the built-in MATLAB function ‘alignsignal’ for this purpose. This function ensures signal synchronization before computing the correlation coefficient, which resulted adequate in our case.
Q1.5 - The shallow neural network architecture includes 20 neurons in hidden layers for regression and 30 for classification. What was the rationale behind choosing these specific numbers of neurons, and were other configurations tested?
A1.5 - Preliminary tests were carried out and the best configurations based on accuracy on prediction (energy regression and damage classification) were selected.
Q1.6 - A 70/30 split was used for training and testing. Was cross-validation employed to ensure the robustness of the model, and if not, what measures were taken to prevent overfitting?
A1.6 – The following sentence has been added for clarification: “The choice of a K-Fold cross validation procedure, considering 5 different combinations of training/test sets with a training/test ratio equal to 70/30 allowed such fitting.”…
Q1.7 - The authors highlight the use of Bayesian Regularization for training. How does this method compare in terms of computational efficiency and accuracy to other training algorithms like Levenberg-Marquardt or Scaled Conjugate Gradient?
A1.7 - Other training algorithms have been tested in both cases providing worst results in terms of accuracy and computational time, therefore have not been discussed in this work.
Q1.8 - Broaden literature review on ML-based damage detection. E.g., Ground penetrating radar-based automated defect identification of bridge decks: A hybrid approach.
A1.8 - Thank you for the valuable suggestion. We have added the recommended work to the reference list and expanded the literature review.
Q1.9 - The manuscript claims the method is computationally efficient. Can the authors provide specific metrics such as training time and prediction time to support this claim, especially for real-time applications?
A1.9 - Thank you for your insightful comment. We have incorporated this important detail into the Conclusions section.
Q1.10 - More future research should be included in conclusion part.
A1.10 - In the Conclusion section, we have added ongoing work regarding interlaboratory comparison, which is an essential activity to improve the robustness of the approach.
Reviewer 2 Report
Comments and Suggestions for Authors
Review Report
sensors-3773713-peer-review-v1
Energy Evaluation and Passive Damage Detection for Structural Health Monitoring in Aerospace Structures Using Machine Learning Models
This manuscript presents a study on structural health monitoring (SHM) of aerospace-grade reinforced panels using machine learning technique. The authors develop and evaluate two neural network-based approaches: one for estimating impact (velocity) energy (via regression), and another for classifying structural damage (via pattern recognition). Experiments were conducted using a flat aluminum plate with riveted stiffeners, where impacts of varying energies were applied at multiple locations. The passive monitoring strategy leverages environmental vibrations (impact events) captured by piezoelectric sensors (PZTs), and features such as time of flight (ToF), RMS, crest factor, and spectral characteristics are extracted for model input. The proposed methodology is implemented using shallow neural networks trained in MATLAB, with results indicating good performance in both impact energy prediction and quadrant-level damage classification.
The overall quality of the manuscript is high, and the reviewer found it engaging. The work merits publication following minor revisions. The comments provided below are intended for the authors’ consideration to further enhance the clarity and completeness of the manuscript.
- Suitability of scope:
- The manuscript is closely aligned with topics such as sensor arrays, AI-enabled sensors application, sensor datasets processing or signal processing for meaningful features extraction. From a scope perspective, it is well suited for publication in the MDPI journal Sensors.
- Originality and Contribution
- The use of shallow neural networks in combination of Bayesian regularization presents an interesting and valuable contribution, as this approach offers a potential solution to the computational tractability issues commonly associated with deep learning methods. The development and validation of such a promising and efficient methodology are commendable, and its publication should be encouraged.
- Methodological Clarifications
- Additional details regarding the flat reinforced plate specimen shown in Figure 1 are necessary to enhance transparency and reproducibility. Specifically, the manuscript should provide information on the rivet specifications, including the rivet type and spacing (rivet interval), as well as whether adhesive was used at the interface between the flat plate and the stringers. If adhesive bonding was employed, the exact product name and manufacturer (or relevant specifications) should be reported.
- Please also provide the specifications (or brand + product number) of the bicomponent glue used to bond the PZTs.
- The description of the impact grid design requires clarification. The current statement—“A regular grid of 11×11 points provides 121 – 4 (sensors) = 117 impact locations, creating a symmetrical pattern, plus 50 impacts randomly chosen to reduce homogeneity and possible overfitting. The resulting 167 (red dots in Figure 1 and Figure 2) impact responses from the plate were…”—is unclear for several reasons. First, the number of grid points described does not appear to correspond to what is shown in Figure 2. Second, the reference to “red dots” in Figures 1 and 2 is problematic, as such markers are not visually identifiable in the figures provided. The authors are encouraged to revise both the textual explanation and the associated figures to ensure consistency and clarity regarding the number, location, and visual representation of the impact points.
- Please provide a clear description of the boundary conditions applied to the specimen plate during impact testing, specifically regarding how the sides of the plate were supported or constrained. Additionally, it would be helpful if the authors could justify whether these boundary conditions are representative of realistic, in-service conditions for aerospace structures. This request is made solely in the interest of transparency; it is fully acceptable if the chosen boundary conditions are not representative of real-world cases, provided this is clearly stated.
- The manuscript should specify the sampling rate used during data acquisition, along with the expected frequency range of the impact signals. The currently provided information—namely, a timebase control of 5 ms/div—is insufficient to assess the adequacy of the sampling rate for capturing the relevant signal characteristics. Without this information, it is difficult to evaluate whether the data acquisition setup is appropriate for the temporal resolution required in this application.
- Although both the mass and diameter of the steel ball are provided in the manuscript, they are presented in separate paragraphs. For improved clarity and ease of reference, it is recommended that both parameters be reported together in the same location, ideally at the first mention of the drop tower setup.
- It is unclear how the authors derived the time of flight (ToF) from the power spectral density (PSD), given that the horizontal axis of the PSD represents frequency rather than time. Could the authors please clarify the procedure or calculation used to obtain ToF from the PSD?
- How many impact height levels were actually used in the experiments? The description of the impact heights and their corresponding velocities is difficult to follow due to scattered and partially overlapping information across different sections. Specifically, Section 2.1 mentions five velocity levels (approximately 2.6, 2.4, 2.2, 2.1, and 1.9 m/s), while two additional velocities (approximately 1.8 and 1.3 m/s, corresponding to half and one-quarter of h_max) are introduced earlier in the same paragraph. This creates confusion, particularly when Section 2.3 refers to seven ball-dropping heights. The authors are encouraged to clearly and explicitly list all seven height levels and their corresponding velocities in a single location to ensure consistency and improve clarity for readers. This clarification remains relevant even though the heights are listed in Table 1.
- The manuscript should clarify how the lower and upper limits of the acceptable velocity prediction range (+/- 10%) presented in Table 1 were defined. What criteria or justification was used to select this specific tolerance range?
- By “Z-Score feature scaling,” do the authors refer to standardization—that is, transforming the data to have zero mean and unit variance? If so, it is recommended to state this explicitly to avoid ambiguity.
- The number “12” shown beneath the Input in Figure 4 suggests that the model uses 12 input features. However, the features are mentioned in a scattered manner throughout Section 2.3, making it difficult to identify all of them. The reviewer was able to locate only eight: (1) Peak, (2) RMS, (3) CF, (4) Correlation coefficient, (5) STFT, (6) PSD, (7) Energy, and (8) ToF. Could the authors please explicitly list all 12 input features in one place for clarity and completeness?
- Many standard model hyperparameters—such as learning rate, dropout rate, and momentum—are not reported in the manuscript. While this may be acceptable given the use of shallow neural networks with Bayesian Regularization in MATLAB (where such parameters are often managed internally or not applicable), it would still be helpful if the authors could explicitly confirm which parameters were fixed by default and which, if any, were manually configured.
- Results and Discussion
- The results are presented in a logical and coherent manner. The authors interpret the findings in the context of relevant literature and refrain from speculative or biased claims. The study’s objectives are fully addressed, and the implications of the results are clearly articulated. One exception is the explanation provided for the reduced classification accuracy in quadrant β. The authors note that “since the artificial damage was caused by manually unriveting the stiffener, probably this less accurate result was a consequence of a more complex kissing bond in the quadrant β case.” This discussion would benefit from further elaboration on how the rivets were removed in the other quadrants, and/or how the formation of kissing bonds in those cases may have been mitigated or avoided. Clarifying this would strengthen the interpretation and provide a more comprehensive understanding of the observed variation.
- Figures, Tables, and Presentation Quality
- Figure 2: Could the authors please clarify the intension of including a QR code in the figure?
- Figure 2 caption: It is recommended to provide a more informative figure caption.
- Could the authors clarify why Figures 6 and 7 are presented separately, despite both showing closely related results that could be more naturally and effectively plotted together in a single figure? If there is no specific justification for separating them, it is recommended to combine the plots to improve visual coherence and facilitate direct comparison.
- The clarity of the confusion matrices for the test dataset and the combined dataset (Figure 9) should be improved. The small font size and visual congestion make it difficult to read the values accurately. Enhancing the resolution, increasing font size, or restructuring the layout would help improve readability.
- It is recommended that a graphical legend be added to Figure 10 to clarify the meaning of the various visual elements, including green dots, red dots, black dots, black-outlined white circles, black boxes, and grey lines. Including a legend would significantly improve the readability and interpretability of the figure.
- English Language and Structure
- The clarity of the following sentence should be improved: “In the case of a real ‘plate assembly’ there would be no quadrants outside the sensors array: each set of 4 PZT would be able to monitor impact phenomena.” It is unclear what is meant by “quadrants” in this context—does this refer to specific sections of the monitored area, or to a spatial division around each sensor array? Furthermore, the phrase “each set of 4 PZT” implies the existence of multiple such sets, while only one set appears to be discussed. If the authors intend to suggest that multiple PZT arrays would be distributed across a target structure in a real case scenario to enable comprehensive monitoring, this should be stated explicitly.
- This sentence needs to be improved for clarity: “Subsequently, half and a quarter of h_max were chosen (v equal about to 1.8 and 1.3 m/s, respectively).”
- The last two paragraphs of Section 2.1 require significant clarification to make the experimental setup understandable. In particular, the description of the damaged scenarios (ii) to (v), involving missing rivets in quadrants α, β, γ, and δ, is ambiguous. It is unclear how the damage was introduced and whether it was cumulative or isolated. For example:
– Were three rivets removed from one quadrant, data acquired, and then an additional three rivets removed from a second quadrant—while retaining the initial damage—thus accumulating damage across quadrants?
– Or were three rivets removed from a single quadrant, data acquired, and then the rivets replaced before moving on to test another quadrant independently?
– Or were the three rivets per quadrant removed incrementally—i.e., one at a time with data collected after each removal—and if so, were rivets replaced before moving on to another quadrant?
This information is essential for understanding the nature and independence of the damage scenarios and for evaluating the classification methodology. The authors are strongly encouraged to revise this section to clearly and explicitly describe the sequence of damage introduction, data acquisition, and whether any rivet replacements occurred between tests.
- The meaning of the (partial) sentence “… adopting a supervised learning approach as suggested in [24], a good choice is represented by a Shallow Neural Network (SNN) made of one or two hidden layers” is unclear and difficult to follow. The authors are encouraged to rephrase this for improved readability. For example, it might be clearer to state: “… adopting a supervised learning approach, as suggested in [24], such as a Shallow Neural Network (SNN) consisting of one or two hidden layers, is considered a suitable choice.” Another example is: “… as suggested in [24], a supervised learning approach, such as a Shallow Neural Network (SNN) consisting of one or two hidden layers, represents a suitable choice.”
- The term “vademecum” is uncommon or used in an unconventional manner in the sentence “The vademecum of the previous flow chart is the following bullet list.” The authors are encouraged to consider rephrasing this for clarity. For example: “The following bullet list summarizes the key steps illustrated in the flow chart above.”
- Keywords, Terminology, and abbreviation
- Please declare the use of abbreviation AS.S.E. Lab to represent AeroSpace Structure Engineering Lab when the abbreviation first appears in the manuscript.
- Is the abbreviation for Crest Factor intended to be CF or CR? Please ensure consistent use of the abbreviation throughout the manuscript.
Author Response
We thank the reviewer for the constructive and encouraging comments. Please find below our point-by-point responses. In the revised manuscript, the changes are highlighted in green for easier verification of the implemented adjustments.
Q2.0 - This manuscript presents a study on structural health monitoring (SHM) of aerospace-grade reinforced panels using machine learning technique. The authors develop and evaluate two neural network-based approaches: one for estimating impact (velocity) energy (via regression), and another for classifying structural damage (via pattern recognition). Experiments were conducted using a flat aluminum plate with riveted stiffeners, where impacts of varying energies were applied at multiple locations. The passive monitoring strategy leverages environmental vibrations (impact events) captured by piezoelectric sensors (PZTs), and features such as time of flight (ToF), RMS, crest factor, and spectral characteristics are extracted for model input. The proposed methodology is implemented using shallow neural networks trained in MATLAB, with results indicating good performance in both impact energy prediction and quadrant-level damage classification.
The overall quality of the manuscript is high, and the reviewer found it engaging. The work merits publication following minor revisions. The comments provided below are intended for the authors’ consideration to further enhance the clarity and completeness of the manuscript.
A2.0 - The authors would like to thank the reviewer for the careful reading of our manuscript and the many insightful remarks. Please, find below the point-by-point list of questions and relative answers.
Suitability of scope:
Q2.1 - The manuscript is closely aligned with topics such as sensor arrays, AI-enabled sensors application, sensor datasets processing or signal processing for meaningful features extraction. From a scope perspective, it is well suited for publication in the MDPI journal Sensors.
A2.1 - We sincerely thank the reviewer for recognizing the suitability of our work for publication in MDPI journal Sensors.
Originality and Contribution
Q2.2 - The use of shallow neural networks in combination of Bayesian regularization presents an interesting and valuable contribution, as this approach offers a potential solution to the computational tractability issues commonly associated with deep learning methods. The development and validation of such a promising and efficient methodology are commendable, and its publication should be encouraged.
A2.2 - Thank you very much for the encouragement. We appreciate your recognition of the development and validation of this promising and efficient methodology. We are actively working on it and believe its publication will contribute significantly to the field.
Methodological Clarifications
Q2.3 - Additional details regarding the flat reinforced plate specimen shown in Figure 1 are necessary to enhance transparency and reproducibility. Specifically, the manuscript should provide information on the rivet specifications, including the rivet type and spacing (rivet interval), as well as whether adhesive was used at the interface between the flat plate and the stringers. If adhesive bonding was employed, the exact product name and manufacturer (or relevant specifications) should be reported.
A2.3 – As per the reviwer’s suggestion, we specified there is no adhesive between the flat plate and the stringers. The rivets characteristics have been included and described in the revised manuscript.
Q2.4 - Please also provide the specifications (or brand + product number) of the bicomponent glue used to bond the PZTs.
A2.4 – Following the reviewer’s comment the specification of the bicomponent glue used has been added, as well.
Q2.5 - The description of the impact grid design requires clarification. The current statement—“A regular grid of 11×11 points provides 121 – 4 (sensors) = 117 impact locations, creating a symmetrical pattern, plus 50 impacts randomly chosen to reduce homogeneity and possible overfitting. The resulting 167 (red dots in Figure 1 and Figure 2) impact responses from the plate were…”—is unclear for several reasons. First, the number of grid points described does not appear to correspond to what is shown in Figure 2. Second, the reference to “red dots” in Figures 1 and 2 is problematic, as such markers are not visually identifiable in the figures provided. The authors are encouraged to revise both the textual explanation and the associated figures to ensure consistency and clarity regarding the number, location, and visual representation of the impact points.
A2.5 - Thank you for highlighting this aspect—your observation regarding the description of the gird design was very helpful. We have modified the figure to enhance clarity, and the caption has been expanded to provide a more detailed explanation.
Q2.6 - Please provide a clear description of the boundary conditions applied to the specimen plate during impact testing, specifically regarding how the sides of the plate were supported or constrained. Additionally, it would be helpful if the authors could justify whether these boundary conditions are representative of realistic, in-service conditions for aerospace structures. This request is made solely in the interest of transparency; it is fully acceptable if the chosen boundary conditions are not representative of real-world cases, provided this is clearly stated.
A2.6 - We thank the reviewer for pointing out this missing detail. The aluminum plate was placed on a soft and homogeneous sponge, as done in previous cited works to ensure proper support and consistency.
Q2.7 - The manuscript should specify the sampling rate used during data acquisition, along with the expected frequency range of the impact signals. The currently provided information—namely, a timebase control of 5 ms/div—is insufficient to assess the adequacy of the sampling rate for capturing the relevant signal characteristics. Without this information, it is difficult to evaluate whether the data acquisition setup is appropriate for the temporal resolution required in this application.
A2.7 - We would like to clarify that the expected frequency range, the 0–40 kHz bandwidth is based on literature values for an aluminum plate 1.1 mm thick, whereas our plate has a thickness of 1.2 mm, thus we considered it a suitable working range. This information has been already explained in the description of the Time of Flight (TOF) evaluation, in Section 2.3 The sampling frequency has now been indicated; its previous omission was an oversight, and we apologize for it. We appreciate the reviewer for having brought this to our attention.
Q2.8 - Although both the mass and diameter of the steel ball are provided in the manuscript, they are presented in separate paragraphs. For improved clarity and ease of reference, it is recommended that both parameters be reported together in the same location, ideally at the first mention of the drop tower setup.
A2.8 - We understand that this may have caused some confusion; therefore, we have moved the specification of the ball mass earlier in the text, placing it alongside the mention of the diameter, immediately following the description of the drop tower.
Q2.9 - It is unclear how the authors derived the time of flight (ToF) from the power spectral density (PSD), given that the horizontal axis of the PSD represents frequency rather than time. Could the authors please clarify the procedure or calculation used to obtain ToF from the PSD?
A2.9 - We thank the reviewer for this important question. Once the Power Spectral Density (PSD) is evaluated, the Time of Flight (ToF) is obtained from the spectrogram by identifying the highest peak corresponding to the frequency of approximately 40 kHz. This approach allows us to extract the ToF despite the PSD’s frequency-domain representation. The procedure has already been described in Section 2.3, and previous work by the authors provides a more detailed explanation. Therefore, in the present manuscript, it has only been briefly recalled, for the sake of conciseness.
Q2.10 - How many impact height levels were actually used in the experiments? The description of the impact heights and their corresponding velocities is difficult to follow due to scattered and partially overlapping information across different sections. Specifically, Section 2.1 mentions five velocity levels (approximately 2.6, 2.4, 2.2, 2.1, and 1.9 m/s), while two additional velocities (approximately 1.8 and 1.3 m/s, corresponding to half and one-quarter of h_max) are introduced earlier in the same paragraph. This creates confusion, particularly when Section 2.3 refers to seven ball-dropping heights. The authors are encouraged to clearly and explicitly list all seven height levels and their corresponding velocities in a single location to ensure consistency and improve clarity for readers. This clarification remains relevant even though the heights are listed in Table 1.
A2.10 - We thank the reviewer for this valuable suggestion. To enhance clarity and facilitate understanding, we have added a bullet-point list in Section 2.1. We believe this formatting improvement makes the information more accessible and easier to follow.
Q2.11 - The manuscript should clarify how the lower and upper limits of the acceptable velocity prediction range (+/- 10%) presented in Table 1 were defined. What criteria or justification was used to select this specific tolerance range?
A2.11 - We thank the reviewer for this important question. The ±10% tolerance range reported in Table 1 was chosen as a practical empirical criterion during preliminary analyses and is supported by the observed consistency of our results. We have clarified this point in the revised manuscript. Additionally, potential measurement uncertainties and their influence on velocity predictions are currently being investigated in ongoing work, which is beyond the scope of the present paper.
Q2.12 - By “Z-Score feature scaling,” do the authors refer to standardization—that is, transforming the data to have zero mean and unit variance? If so, it is recommended to state this explicitly to avoid ambiguity.
A2.12 – We thank the reviewer and confirm that this is indeed the case: Z-score feature scaling provides a standardized dataset that has a mean of 0 and a standard deviation of 1, while preserving the shape characteristics of the original distribution, including skewness and kurtosis. For clarification, the following remark has been added to the manuscript: “Z-score features scaling (standardized dataset to have zero mean and unit variance)”.
Q2.13 - The number “12” shown beneath the Input in Figure 4 suggests that the model uses 12 input features. However, the features are mentioned in a scattered manner throughout Section 2.3, making it difficult to identify all of them. The reviewer was able to locate only eight: (1) Peak, (2) RMS, (3) CF, (4) Correlation coefficient, (5) STFT, (6) PSD, (7) Energy, and (8) ToF. Could the authors please explicitly list all 12 input features in one place for clarity and completeness?
A2.13 - Thank you for having highlighted this missing explanation. We have now clarified in the manuscript that the feature set used for energy-related analysis includes 4 TOF values, 4 peak values, and 4 RMS values. For damage detection, the same 12 features are used, with the addition of the correlation coefficient—one per sensor, comparing the signals from the undamaged and damaged plate—resulting in a total of 16 features.
Q2.14 - Many standard model hyperparameters—such as learning rate, dropout rate, and momentum—are not reported in the manuscript. While this may be acceptable given the use of shallow neural networks with Bayesian Regularization in MATLAB (where such parameters are often managed internally or not applicable), it would still be helpful if the authors could explicitly confirm which parameters were fixed by default and which, if any, were manually configured.
A2.14 - The parameters manually configured are those reported in the manuscript. The other ones were managed internally or not applicable.
Results and Discussion
Q2.15 - The results are presented in a logical and coherent manner. The authors interpret the findings in the context of relevant literature and refrain from speculative or biased claims. The study’s objectives are fully addressed, and the implications of the results are clearly articulated. One exception is the explanation provided for the reduced classification accuracy in quadrant β. The authors note that “since the artificial damage was caused by manually unriveting the stiffener, probably this less accurate result was a consequence of a more complex kissing bond in the quadrant β case.” This discussion would benefit from further elaboration on how the rivets were removed in the other quadrants, and/or how the formation of kissing bonds in those cases may have been mitigated or avoided. Clarifying this would strengthen the interpretation and provide a more comprehensive understanding of the observed variation.
A2.15 - We thank the reviewer for the observation. While this situation represents an anomaly caused by the manual operation used to remove the stiffener, which inherently suffers from limited repeatability, we agree that implementing a more stable and controlled system for generating impacts would be beneficial. However, this lies beyond the scope of the present study and is therefore discussed as a limitation and a direction for future work.
Figures, Tables, and Presentation Quality
Q2.16 - Figure 2: Could the authors please clarify the intension of including a QR code in the figure?
A2.16 - Done, as requested—thank you for the suggestion.
Q2.17 - Figure 2 caption: It is recommended to provide a more informative figure caption.
A2.17 - Implemented as suggested. Thank you.
Q2.18 - Could the authors clarify why Figures 6 and 7 are presented separately, despite both showing closely related results that could be more naturally and effectively plotted together in a single figure? If there is no specific justification for separating them, it is recommended to combine the plots to improve visual coherence and facilitate direct comparison.
A2.18 - The change has been made, many thanks.
Q2.19 - The clarity of the confusion matrices for the test dataset and the combined dataset (Figure 9) should be improved. The small font size and visual congestion make it difficult to read the values accurately. Enhancing the resolution, increasing font size, or restructuring the layout would help improve readability.
A2.19 - This has now been addressed, thank you.
Q2.20 - It is recommended that a graphical legend be added to Figure 10 to clarify the meaning of the various visual elements, including green dots, red dots, black dots, black-outlined white circles, black boxes, and grey lines. Including a legend would significantly improve the readability and interpretability of the figure.
A2.20 - This has been done, thank you.
English Language and Structure
Q2.21 - The clarity of the following sentence should be improved: “In the case of a real ‘plate assembly’ there would be no quadrants outside the sensors array: each set of 4 PZT would be able to monitor impact phenomena.” It is unclear what is meant by “quadrants” in this context—does this refer to specific sections of the monitored area, or to a spatial division around each sensor array? Furthermore, the phrase “each set of 4 PZT” implies the existence of multiple such sets, while only one set appears to be discussed. If the authors intend to suggest that multiple PZT arrays would be distributed across a target structure in a real case scenario to enable comprehensive monitoring, this should be stated explicitly.
A2.21 - According to the reviewer’s suggestion, we have improved Figure 2, which now provides a clearer and more illustrative representation of the concept.
Q.2.22 - This sentence needs to be improved for clarity: “Subsequently, half and a quarter of h_max were chosen (v equal about to 1.8 and 1.3 m/s, respectively).”
A2.22 – The sentence has been deleted, and the corresponding information has been reorganized into a bullet-point list for greater clarity, summarizing the various conditions under investigation.
Q2.23 - The last two paragraphs of Section 2.1 require significant clarification to make the experimental setup understandable. In particular, the description of the damaged scenarios (ii) to (v), involving missing rivets in quadrants α, β, γ, and δ, is ambiguous. It is unclear how the damage was introduced and whether it was cumulative or isolated. For example:
– Were three rivets removed from one quadrant, data acquired, and then an additional three rivets removed from a second quadrant—while retaining the initial damage—thus accumulating damage across quadrants?
– Or were three rivets removed from a single quadrant, data acquired, and then the rivets replaced before moving on to test another quadrant independently?
– Or were the three rivets per quadrant removed incrementally—i.e., one at a time with data collected after each removal—and if so, were rivets replaced before moving on to another quadrant?
This information is essential for understanding the nature and independence of the damage scenarios and for evaluating the classification methodology. The authors are strongly encouraged to revise this section to clearly and explicitly describe the sequence of damage introduction, data acquisition, and whether any rivet replacements occurred between tests.
A2.23 - We confirm that the situation corresponds to one of those described by the Reviewer: three rivets were removed from a single quadrant, data were acquired, and then the rivets were reinserted before proceeding to test another quadrant independently.
Q2.24 - The meaning of the (partial) sentence “… adopting a supervised learning approach as suggested in [24], a good choice is represented by a Shallow Neural Network (SNN) made of one or two hidden layers” is unclear and difficult to follow. The authors are encouraged to rephrase this for improved readability. For example, it might be clearer to state: “… adopting a supervised learning approach, as suggested in [24], such as a Shallow Neural Network (SNN) consisting of one or two hidden layers, is considered a suitable choice.” Another example is: “… as suggested in [24], a supervised learning approach, such as a Shallow Neural Network (SNN) consisting of one or two hidden layers, represents a suitable choice.”
A2.24 - We appreciate the reviewer’s suggestion and have implemented it in the revised version of the manuscript.
Q2.25 - The term “vademecum” is uncommon or used in an unconventional manner in the sentence “The vademecum of the previous flow chart is the following bullet list.” The authors are encouraged to consider rephrasing this for clarity. For example: “The following bullet list summarizes the key steps illustrated in the flow chart above.”
A2.25 – The sentence has been changed into: “The following bullet list provides a practical guideline that summarises the key steps of the approach illustrated in the flow chart”.
Keywords, Terminology, and abbreviation
Q2.26 - Please declare the use of abbreviation AS.S.E. Lab to represent AeroSpace Structure Engineering Lab when the abbreviation first appears in the manuscript.
A2.26 - Thank you for the suggestion, it has been done, as requested.
Q2.27 - Is the abbreviation for Crest Factor intended to be CF or CR? Please ensure consistent use of the abbreviation throughout the manuscript.
A2.27 - We are grateful to the reviewer for pointing this out. The terms 'CR' and ‘CF’ were both removed, as Crest Factor has not been used for any of the algorithms implemented in the present study.
Reviewer 3 Report
Comments and Suggestions for Authors
Please check the attachment

Author Response
We thank the reviewer for the constructive and encouraging comments. Please find below our point-by-point responses. In the revised manuscript, the changes are highlighted in cyan for easier verification of the implemented adjustments.
This paper describes an experimental study to investigate the role of ML in identifying and characterizing the structural effects of impacts on an aerospace aluminum panel. Although the experimental methodology for data generation, and use of ML for SHM in aerospace is now relevant, the paper needs significant revisions in the present form as outlined next:
Q3.1 - Besides saying aerospace aluminum panel and piezo sensors, I feel that you require a complete and accurate description of the experimental procedure. For instance, you could clearly provide a description of how the impacts were generated whether through instrumented impact, drop hammer, etc.
A3.1 – We thank the reviewer for this insightful suggestion. While we agree it is valuable, we believe that the relevant information about the experiments is already included in Section 2.1 (lines 153-158)
Q3.2 - Provide the energy levels or force profiles of the impacts, the actual impact locations on the panel, and how you documented that each of the impacts was applied consistently and uniformly across the experiments.
A3.2 - Thank you; we have clarified this point with an additional sentence in the manuscript.
Q3.3 - The model and technical specifications of the data acquisition (DAQ) system should have been specified along with the sampling rate, duration of data acquisition for each impact event, and if any initial signal conditioning was done.
A3.3 - The oscilloscope model was already specified in the manuscript. Regarding the initial signal conditioning, a trigger was used, and this has now been explicitly clarified in the revised version. Sampling rate and duration of each impact event have also been added to the textual description.
Q3.4 - Please include more granular information about the produced impact datasets, such as the number of impacts or samples obtained for each of the severity levels and how many data events were assigned to the training, validation, and test sets. In addition, please discuss each of the signal preprocessing steps that were performed on the raw vibrational data (noise filtering, signal normalization, time-windowing, feature extraction etc.).
A3.4 - Training and testing were performed using a 70%-30% data split, as already highlighted in the previous version of the manuscript. Moreover, the algorithm processes the raw signal after applying a second-order high-pass Butterworth filter. A Hamming time-window was then applied for the computation of the Power Spectral Density (PSD). Finally, the signal was normalized using z-score scaling during the training phase. The above-mentioned elements have been more clearly highlighted in the revised manuscript to enhance understanding and ensure completeness.
Q3.5 - The authors depended on a Shallow neural network, which is very general. Please provide the actual configuration of the architecture, including the number of hidden layers, number of neurons in each layer,
the activation functions used at each layer, and the output layer configuration as appropriate for the task being performed. Please identify specific training parameters, including the optimizer algorithm (Adam, SGD, etc.), learning rate, batch size, number of epochs, and all regularization methods (e.g. dropout rates, L1/L2 regularization coefficients).
A3.5 - Figures 4 and 8, along with the preceding bullet points, provide all the relevant characteristics, as well as additional parameters that were left unspecified. In particular, the parameters manually configured are those reported in the manuscript. The other ones were managed internally or their fine-tuning not applicable.
Q3.6 - To credibly demonstrate the merit of the proposed machine learning approach, please try to compare the approach against relevant baseline techniques directly and fairly.
A3.6 - With reference to the previous work of the Authors on energy classification, we have improved the approach by implementing in the present manuscript a regression analysis.
Q3.7 - Pleas further improve the discussion and future work by stating the major limitations and areas of improvement.
A3.7 - Improvements have already been discussed extensively in the Conclusions section. Some enhancements related to validation are currently underway. Limitations remain regarding the impact magnitude (energy), which are primarily due to the performance constraints of the acquisition system in avoiding signal saturation.
Reviewer 4 Report
Comments and Suggestions for Authors
Please see the attachment.

Author Response
We thank the reviewer for the constructive and encouraging comments. Please find below our point-by-point responses. In the revised manuscript, the changes are highlighted in red for easier verification of the implemented adjustments.
The manuscript presents the estimation of impact energy and the location of preexisting damages using shallow neural networks. An accepted prediction accuracy was presented. The paper is overall well-written. However, the following concerns should be addressed before the manuscript can be accepted for publication:
Q4.1 - The manuscript focuses on the estimation of impact energy, but the location of the impact is also very important when analyzing the impact event. I suggest the authors should mention this aspect in the context.
A4.1 – Thank you for your interest in this topic, which has been the subject of another study. For this reason, we have cited that work in the present manuscript.
Q4.2 - I suggest the authors show some example signals under different impact energies and some example signals received from the same transducer under the same impact location with the same impact energy but for different damage quadrant. In addition, the corresponding description and discussion about these signals should be added as well.
A4.2 – Examples have been added in accordance with the reviewer’s suggestions. However, we have limited the number of figures compared to those requested, as it can be observed that additional figures do not provide significant new evidence to the naked eye without the further processing proposed in our work.
Q4.3 - It is unclear whether the features ToF, RMS, max peak, CF, and correlation coefficients are all used for both impact energy estimation and damage localization? Following this question, the authors need to indicate what the 12 and 16 inputs composed of for these two studies, respectively.
A4.3 - Thank you for having highlighted this missing explanation. We have now clarified in the manuscript that the feature set used for energy-related analysis includes 4 TOF values, 4 peak values, and 4 RMS values. For damage detection, the same 12 features are used, with the addition of the correlation coefficient—one per sensor, comparing the signals from the undamaged and damaged plate—resulting in a total of 16 features.
Q4.4 - Is it needed to separately plot Figures 6 and 7? Since they are showing all the cases of different heights, there is no need to plot them in this way.
A4.4 – The reviewer’s comment regarding visualization has been taken into account, and the sub-figures are now presented together in Figure 7.
Q4.5 - In Figure 9, the resolution of the test confusion matrix and all confusion matrix is very low. The authors should improve the picture quality.
A4.5 - This has now been addressed, thank you.
Q4.6 - In Figure 11, all the curves for the training and all ROC almost overlap with each other. It is very difficult to distinguish their individual trends.
A4.6 - We thank the reviewer for the observation. The ROC curves appear in this way because the training results are very close to the upper-left corner, indicating high sensitivity and specificity, with only two false negatives. Test ROC curves also remain well above the line of no-discrimination. These results reflect the robustness of the proposed method, as also confirmed by AUC values consistently above 0.9. A clarifying paragraph has been added to the manuscript accordingly.
Round 2
Reviewer 1 Report
Comments and Suggestions for Authors
All the technical issues have been addressed by the authors. I suggest this revised version can be accepted for publication.
Author Response
We are thankful for the reviewer’s time and effort in reviewing our manuscript.Reviewer 3 Report
Comments and Suggestions for Authors
No further comments
Author Response
We are thankful for the reviewer’s time and effort in reviewing our manuscript.Reviewer 4 Report
Comments and Suggestions for Authors
The manuscript has been improved after the revision. One minor issue is the caption of Figure 12, where the word "Damaje" should be "Damage".
Author Response
We are thankful for the reviewer’s time and effort in reviewing our manuscript and for drawing our attention to the oversight, which has now been corrected.